# The Current Status and Future Perspectives of Beta-Lactam Therapeutic Drug Monitoring in Critically Ill Patients

**DOI:** 10.3390/antibiotics12040681

**Published:** 2023-03-30

**Authors:** Emmanuel Novy, Hugo Martinière, Claire Roger

**Affiliations:** 1Department of Anesthesiology and Critical Care Medicine, Institut Lorrain du Coeur Et Des Vaisseaux, University Hospital of Nancy, Rue du Morvan, 54511 Vandoeuvre-les Nancy, France; 2SIMPA, UR 7300, Faculté de Médecine, Maïeutique et Métiers de la Santé, Campus Brabois Santé, University of Lorraine, 54000 Nancy, France; 3Department of Anesthesiology and Intensive Care, Pain and Emergency Medicine, Nimes-Caremeau University Hospital, Place du Professeur Robert Debré, CEDEX 09, 30029 Nimes, France; 4UR UM 103 IMAGINE, Faculty of Medicine, Montpellier University, 30029 Nimes, France

**Keywords:** beta-lactam, therapeutic drug monitoring, ICU, PK/PD target, PK/PD software

## Abstract

Beta-lactams (BL) are the first line agents for the antibiotic management of critically ill patients with sepsis or septic shock. BL are hydrophilic antibiotics particularly subject to unpredictable concentrations in the context of critical illness because of pharmacokinetic (PK) and pharmacodynamics (PD) alterations. Thus, during the last decade, the literature focusing on the interest of BL therapeutic drug monitoring (TDM) in the intensive care unit (ICU) setting has been exponential. Moreover, recent guidelines strongly encourage to optimize BL therapy using a PK/PD approach with TDM. Unfortunately, several barriers exist regarding TDM access and interpretation. Consequently, adherence to routine TDM in ICU remains quite low. Lastly, recent clinical studies failed to demonstrate any improvement in mortality with the use of TDM in ICU patients. This review will first aim at explaining the value and complexity of the TDM process when translating it to critically ill patient bedside management, interpretating the results of clinical studies and discussion of the points which need to be addressed before conducting further TDM studies on clinical outcomes. In a second time, this review will focus on the future aspects of TDM integrating toxicodynamics, model informed precision dosing (MIPD) and “at risk” ICU populations that deserve further investigations to demonstrate positive clinical outcomes.

## 1. Interest of Beta-Lactam Therapeutic Drug Monitoring in the Critically Ill Patient

Sepsis and septic shock are one of the most common causes of intensive care unit (ICU) admission and are associated with increased mortality [1]. The scarcity of new antibiotics that arrives in the pipeline to counter the growing concern of antimicrobial resistance led the medical community to better use the current antibiotics available [2].

Beta-lactams (BL) are the most prescribed antibiotic in critically ill patients. BL are hydrophilic antibiotics particularly subjects to unpredictable concentrations in the context of critical illness due to pharmacokinetic (PK) and pharmacodynamic (PD) alterations [3]. Basically, the two main pathophysiological changes are an increased volume of distribution and modified renal and/or hepatic clearance [4]. In addition to dynamic changes in physiological function, ICU therapies such as massive vascular filling and supportive extracorporeal therapies, such as renal replacement therapy and extracorporeal membrane oxygenation, also impact antibiotic concentrations [5]. The highest risk at the early stage of sepsis remains BL underexposure [6]. It has been demonstrated that ICU patients with BL underexposure have a 1.5-fold higher risk of clinical failure and need antibiotic escalation and death [7]. Regarding the therapeutic target, ICU patients frequently faced high-resistance bacteria with higher MICs than in non-ICU patients [8]. Because of the high risk of underdosage and the highest required PD target, optimization of the BL dose in critically ill patients appears essential. Moreover, pathophysiological alterations are a dynamic process in ICU patients and could change every day. From a PK point of view, at the onset of the septic shock, there is high risk of BL underdosing. From a PD point of view, during this early phase, infections are frequently undocumented or documented but without any minimal inhibitory concentration (MIC) determination. Consequently, empirical PK/PD target are mostly based on a “worst-case scenario” and are generally high. Subsequently, PK alterations could be variable, depending on the severity of the septic shock, from renal insufficiency requiring renal replacement therapy to normal renal function for instance. At this stage, infection is generally documented, and MIC could be available. Thus, PK/PD target should be reevaluated. The Figure 1 depicts one example of timeline regarding both PK and PD alterations during the antibiotic course of critically ill patient.

Lastly, recent guidelines strongly encourage optimizing antibiotic prescription based on their PK/PD properties and using therapeutic drug monitoring (TDM) to ensure adequate dose and concentrations [9,10]. 

## 2. Barriers to Overcome to Increase BL Therapeutic Drug Monitoring Adherence

According to Watson et al., “Therapeutic Drug Monitoring (TDM) is a measurement made in the laboratory of a parameter which, with appropriate interpretation, will directly influence prescribing procedures” [11]. This definition implies a multidisciplinary approach from the antibiotic concentration measurement (laboratory level) to its interpretation in the clinical context (clinician level). It also supposed a direct link between drug concentration and therapeutic effect (efficacy) and/or toxic effect (toxicity). One last basic principle for TDM concerns the site of measure, mostly the plasma where concentrations are generally higher than in the tissue of interest [12].

The two antibiotic classes that better illustrate TDM principles are aminoglycosides and vancomycin [13]. Indeed, these two antibiotic classes have a narrow therapeutic index and a high risk of toxicity. However, in both cases, toxic thresholds to avoid toxicity and adequate concentrations to ensure efficacy are well known in the literature [14,15,16]. Moreover, concerning TDM access, both classes are usually analyzed using immunoassays that are relatively cheap, not time consuming, not requiring well-trained personal and with good reproducibility [16]. This allows the results of measurement to be rapidly available for clinicians. Lastly, TDM of these two antibiotics are available worldwide and 24/7. Recent surveys showed that TDM for these two classes of antibiotic is well implemented [17,18].

Contrary to aminoglycosides and vancomycin, BL TDM is more challenging for several reasons [19]. The first reason is related to the physicochemical properties of the drug itself. BL are hydrophilic drugs with low to moderate protein binding (except for ceftriaxone and flucloxacillin which exhibit high protein binding) [20]. The most binding protein is albumin. This is important to emphasize for BL TDM interpretation, because only the free fraction of the drug is active. In the critically ill patient, hypoalbuminemia frequently occurs [21]. Unfortunately, most of centers where TDM is available measure the total concentration of BL [22]. In this case, to appropriately interpret the results of BL concentrations, clinicians should correct these concentrations by the free fraction based on the percentage of protein binding published in the literature [23]. Secondly, the optimal PK/PD target remains debated. In critically ill patients, 100% of the time where the free concentration is above the MIC (100% fT > MIC) is often suggested as a therapeutic target [7,10]. Some authors also proposed, in addition to the time spend above the MIC, to add a multiple of the MIC (i.e., 2–5 × MIC) [24,25,26]. The rationale to increase the ratio of MIC target is based on tissue diffusion, technical issues about the uncertainty of MIC and BL measures and resistance suppression [23,27]. However, these propositions come mostly from observational studies showing a better microbiological cure and clinical cure without any impact on the mortality. More research is warranted to determine whether the target should be increased from 100% fT > MIC to 100% fT > *k* × MIC (where *k* is a multiple factor strictly above 1). Finally, most of the literature about BL PK/PD target concerns ICU patients infected with bacteria expressing high MICs or target those based on a “worst-case” scenario to justify high doses. No study focused on dose and PK/PD target for the treatment of low MIC bacteria, which account for most patients, including ICU patients [28]. In documented infections, the greatest issue regarding MIC remains the turnaround time. Indeed, 24 h minimum are required to obtain a positive culture, and, depending on the MIC determination method, an additional 24 h may be necessary [29]. This timeline cannot be easily reduced, and an alternative must be taken, such as epidemiological cutoff (ECOFF) based on local ecology.

Secondly, unlike assays for aminoglycosides and vancomycin, no commercially available assay for BL routine monitoring has been implemented [30]. Most centers use in-house methods for BL quantification that utilize chromatographic separation coupled to ultraviolet (UV) or mass spectrometric (MS) detection. Having in-house high-performance liquid chromatography (HPLC) or MS requires expensive equipment and staff expertise that limit its deployment. Moreover, BL are instable at room temperature, and samples should be maintained on ice or frozen until processed to limit degradation. Recent surveys have highlighted that most centers have to export their analyses leading to delays of results and risk of sample degradation [22]. These pre-analytical issues added to the questions on the optimal PK/PD target for BL make BL a class apart.

Table 1 summarizes points to optimize before TDM implementation.

## 3. Clinical Evidence Supporting BL TDM

During the year 2022, four systematic reviews and meta-analysis have been published about the interest of BL TDM in critically ill patients [19,28,31,32]. Two were focused on the effect of BL TDM on clinical outcome such as mortality and emergence of antimicrobial resistance [31,32]. These two reviews have analyzed 39 studies knowing that some studies were included in both reviews. None of these two reviews have showed an association between BL TDM use and mortality or emergence of antimicrobial resistance. Moreover, according to the authors, there was a high risk of bias (level critical—serious) on items such as “deviation from intended intervention” and “confounding”. These two biases could have significantly influenced the interest of the TDM itself (i.e., when not following the TDM indications) and the impact on mortality with the confounding factors.

In both reviews, there were few prospective randomized clinical trials (RCT) [33,34,35]. Three RCTs were present in the two reviews and totalize 111 patients of various origins: septic patients with normal renal function (n = 41 [35]), neutropenic patients (n = 32 [33]) and burns patients (n = 38 [34]). Therefore, they included highly specific sub population of ICU patients which could question the extrapolation of the results in the general ICU population. Patients with renal replacement therapy (RRT) or augmented renal clearance (ARC), two major sources of BL concentration modifications, were not included. In neutropenic patients, general PK alteration frequently occurred, and, most importantly, the absence of neutrophils altered the bacterial clearance [36] and, thus, changed the PK/PD target to attain. Lastly, burns patients constitute a specific population about their own PK alterations and specific infectious risk because of the loss of protective integument [37]. 

The most recent RCT about BL TDM use and its impact on mortality was the study from Hagel et al. [38] which was included in the review from Mangalore [32]. This RCT has enrolled the largest sample size (n = 249) and was multicenter, and the population was the most representative of the general ICU population (i.e., 74% of patients had septic shock; the median SOFA score at admission was 12.1 ± 2.8). Only the piperacillin (in piperacillin/tazobactam combination) was evaluated, and the PK/PD target was based on the MIC of *Pseudomonas aeruginosa* for empirical therapy. The trial was negative on its primary outcome (difference in mean SOFA score with TDM vs. no TDM, *p* = 0.39). A 4.2% lower mortality and a higher rate of microbiological and clinical cure were observed, but without reaching statistical significance. The rate of target attainment was better with the use of TDM with less underdosing occurrence. Nevertheless, when focusing on target attainment, less than 50% of patients reached the optimal target (without any overdosing) within the five first days with a nadir during the first day. According to the authors, the negative results on mortality could be explained by the high PK/PD target. Indeed, on day one, the PK/PD target was based on MIC from *Pseudomonas aeruginosa* namely 16 mg/L for piperacillin leading to 19% (control group) and 33.9% (TDM group) attainment. However, as most of the identified bacteria displayed a lowest MIC, this non-target attainment could not result in poor outcomes. They are few studies comparing an a priori PK/PD target attainment (based on worst-case scenario with high MIC mostly for empirical therapy) and a posteriori MIC, after bacteria identification. One study from Leon et al., showed in critically ill patients with intra-abdominal infections requiring surgery an increase in PK/PD target attainment before (33%) and after documentation (71%) [39]. This point highlights the crucial role of the MIC used for the choice of the PK/PD target. 

The two other reviews depicted the challenge of clinical studies about BL PK/PD target and the role of TDM [19,28]. They both highlighted two main challenges: (i)The choice of the optimal PK/PD target is unknown and depends on the administration method of the considered BL.(ii)The choice of the MIC for the target (“true MIC” or ECOFF-based) and its determination (risk of interstrain and interlaboratory differences) [40].

The recently published study from Magreault and colleagues summarized the points to consider for MIC interpretation. They reviewed MIC microbiological indications and integrated the MIC in a microbiological, pharmacological and clinical situation approach [41]. One aspect which differs in clinical studies from daily routine practice is the delay for TDM results. Whatever the study, TDM results were available a few hours after the samples. One recent survey about routine practice of TDM highlighted the delay response time as a major barrier for TDM implementation [22]. Moreover, interpretation of TDM results was well known by the prescribers because of the available algorithm to adjust the dose, and the included centers were experts in BL TDM. The same survey about routine practice of TDM showed the difficulty regarding TDM results interpretation as the second major barrier, especially in a non-expert center. All in all, reduction in the delay for TDM results and help for interpretation are the two main objectives to translate results from clinical study into the daily routine practice. Figure 1 depicts all the challenges faced by clinical trials aiming at evaluating the interest of TDM on PK/PD target attainment and its impact on clinical outcomes.

Table 2 proposed a new critical review of the four most recent RCTs on BL TDM use and its impact on mortality.

A focus on PK/PD target according to the choice of the MIC and TDM management was highlighted. Several concerns could be addressed:(i)The heterogeneity of included population with a lot of confounding factors.(ii)An imbalance between a priori target (worst-case scenario/MIC of *Pseudomonas aeruginosa* for empirical treatment) and the actual MIC after bacterial documentation (MIC much lower) without PK/PD target adjustment.(iii)The variability in algorithm for dose adjustment.(iv)The approximation of the renal function measurement leading to possible overestimation of the true renal function, which is a confounding factor especially for BL.(v)The most used concentration of BL was the total fraction, although the PD target was based on the free concentration which needs an estimation.

These concerns should probably partly explain the negative results. Moreover, we must emphasize that, due the interventional design of the study, TDM results were available rapidly, which is not the case for most of centers. Finally, if piperacillin with tazobactam was the most evaluated BL, we need to be cautious about the tazobactam PK/PD changes. According to the review from Monogue [42], there is a lack of data about PK/PD of beta-lactamase inhibitors regarding, among others, the threshold for efficacy for instance. The role of the beta-lactamase inhibitors for the treatment of multidrug-resistant bacteria is crucial and, thus, probably influences the results like the microbiological cure and ultimately, the mortality.

Based on all these challenges, we suggest five key points for designing further studies aiming at assessing the interest of TDM to decrease mortality in ICU patients:To evaluate one BL per study: dose and administration method as well as algorithm of dose adjustment must be described.To avoid dual therapy to limit confounder: if required, in case of septic shock for instance, the choice of the second antibiotic must be unique with adequate dose and the statistical analysis must consider this variable for the final interpretation.To include one origin of infection ideally or infection with or without a source control but not both: indeed, when a source control is possible (i.e., catheter removal, surgical lavage, etc.) the clinical and microbiological cure would depend on the antibiotic adequate concentration and the reduced bacterial load allowed by an external factor. Thus, to include pneumonia and peritonitis for instance could alter comparability of patients.To include a homogeneous population of ICU patients to avoid confounding factors: If immunocompromised patients are evaluated, it must be an inclusion criterion. Indeed, as mentioned earlier, the absence of neutrophils in neutropenic patients altered the bacterial clearance that could compromise the microbiological cure [36]. Another frequent confounding factor is renal failure [43] or ARC [44]. These phenomena both frequently occur in ICU patients and should be considered. Nevertheless, it is a pity that almost all PK/PD studies published nowadays used estimation of the renal function with the CKD-EPI or Cockcroft–Gault formula. Indeed, it has been strongly demonstrated that these two formulae are less accurate in critically ill patients compared to the measured renal clearance using urinary collection [45,46].To carefully choose the MIC used for PK/PD target: most studies based the PK/PD target attainment on day 1 (empirical phase) on a worst-case scenario considering the MIC of a low-susceptible pathogen, mostly *Pseudomonas aeruginosa.* Whatever the choice of the MIC for the empirical phase, it must be reconsidered with the documented MIC, a posteriori or based on the local ecology.

Figure 2 lists the point to consider when constructing future studies on PK/PD target attainment (2A, upper panel) and summarizes the five proposed tips (2B, lower panel) 

## 4. Beta-Lactam Therapeutic Drug Monitoring in Critically Ill Patients: Time to Consider Toxicity? Role of TDM for Assessing BL Toxicity

A growing body of evidence indicates that BL may cause significant toxicity in specific populations, such as the ICU population [47]. Therefore, TDM is also essential in preventing unnecessary excessively high BL exposure that may lead to toxicities. As the threshold BL concentrations for dose-dependent toxicity are generally high, this allows the use of higher initial empirical dosing regimens that can be subsequently refined with TDM. Recently, some studies have demonstrated the role of TDM to minimize non-BL antimicrobial-related toxicity [48]. In a retrospective study including 93 patients, no excessive drug toxicity associated with higher than licensed doses based on TDM was found for either meropenem or piperacillin tazobactam although mean daily doses were more than 40% higher in the high-dose groups [49]. However, the main barrier to TDM-based dosing adjustment to limit toxicity is the lack of well-established thresholds for BL, and efforts to determine toxicodynamics targets are strongly needed. Some studies have focused on the concentration-neurotoxicity relationship of BL in the intensive care setting. Cefepime trough concentrations above 22 mg/L (when administered by intermittent infusions) or concentrations at steady state above 35 mg/L (when administered by continuous infusion) have been associated with neurotoxicity in 50% of patients [50,51]. Comparatively, the same risk has been reported for trough above 64 mg/L for meropenem, 125 mg/L for flucloxacillin and 360 mg/L for piperacillin (used without tazobactam) [52]. In combination with tazobactam, a plasma steady-state concentration of piperacillin above 157 mg/L is predictive of the occurrence of neurological disorders in ICU patients with a specificity of 97% and a sensitivity of 52% [53]. Finally, when the fCmin normalized to the EUCAST clinical breakpoint for *Pseudomonas aeruginosa* (i.e., fCmin/MIC P. *aeruginosa* ratio) exceeded 8, a significant deterioration of the neurological status occurred in approximately half of the ICU patients treated with piperacillin/tazobactam and approximately two-thirds of the ICU patients treated with meropenem [54]. It is of note that an ongoing prospective clinical trial, the OPTIMAL TDM Study (NCT03790631), aims to address the toxicity thresholds of cefepime, imipenem, meropenem, piperacillin, flucloxacillin, amoxicillin and ceftazidime. However, the impact of TDM guided dosing adjustment to prevent harmful BL concentrations and to improve clinical outcome has yet to be determined. 

## 5. Beta-Lactam Therapeutic Drug Monitoring in Critically Ill Patients: Time to Integrate Antimicrobial PK/PD Software?

As seen previously, the traditional approaches of TDM have some limitations. TDM relies on manual data entry, plasma-level determination and interpretation by pharmacists or physicians with PK/PD background. Finally, TDM dosing guidance may become available after several doses, while adequate exposure may be needed right from the start of therapy. To overcome the barriers related to antimicrobial TDM-guided dose optimization, innovative approaches using health record data in real time have been proposed (Figure 3).

A computerized approach using dedicated software that can integrate in their algorithm the different variables affecting antimicrobial PK such as weight, renal function or age could allow a more precise dose adaptation [55]. Model informed precision dosing (MIPD) uses a mathematical model to interpret the measured drug concentration. Thus, MIPD is data driven, relying on the individual patient’s current characteristics and clinical data and drug properties. To integrate the necessary data to inform the modelling, interfacing with health record data in real time is crucial. However, MIPD generally requires custom-made software tools as generic modelling software are too complex for clinicians to apply. Until recently, very few studies have focused on individualized and computerized dose adaptation to optimize antimicrobial therapy in ICU patients [56,57,58,59,60,61]. These approaches aim to better understand each patient’s unique pharmacological profile. 

### 5.1. Dosing Software Principles

Recently, complex mathematical modeling and PK models have been embedded into dosing software to assist with drug dosing. According to the model used in the computer program, three categories of dosing software are currently used: (a)Linear regression model.

The simplest dosing software relies on linear regression model. This method uses a posteriori drug dosing calculations where the patient’s pharmacokinetic parameters are calculated from at least two measured serum concentrations and assume a one compartment model. Based on the PK results, the program will determine the most appropriate dosing regimen for the patient [62]. Several pitfalls of linear regression models have been identified. This approach does not consider a specific PK population and does not predict an initial dose, and each analysis is performed independently without considering any change in the patient’s characteristics over the course of time. 

(b)Population PK-based dosing software.

This approach can be compared to an improved nomogram. Unlike the linear regression model, a single measurement is sufficient to generate dose predictions. This method does not consider any change in patient parameters and does not adapt to the previous determinations. The dose recommendation is only based on population PK parameters without using the patient individual PK results. This approach is considered as an a priori dosing method. This limited view results in a likely loss of reliability in patients whose parameters may change significantly during treatment [63,64]. 

(c)Bayesian forecasting software.

Dosing software implementing Bayesian method combines a given population PK model with the data from an individual patient to determine optimal dose adjustment. Population PK data are used to a priori determine the recommended dose likely to achieve a predefined PK/PD target. When measured drug concentrations become available, these data together with existing population PK data can be used to derive the individual PK parameters using Bayesian estimation and predict individualized dosing regimen. The individual PK parameter estimate is referred to as the maximum a posteriori (MAP) Bayesian estimate. These parameters are derived from the characteristics of the patient for whom the optimal dose is required and will therefore have an important influence on the final simulation. Thus, Bayesian forecasting presents the advantage of using TDM data that strengthen the accuracy of dosing recommendations and consider inter-individual variability. The Bayesian approach offers the advantage that it makes optimal use of all information contained in the population model (a priori) combined with the most current pharmacokinetic information from the patient (a posteriori) to develop the patient’s most precise regimen [65]. Each new serum level collected helps improve the modeling of a patient’s unique PK parameters, further improving dosing recommendations and predictions. This approach can be applied under complicated dosing regimens and non–steady-state conditions and using single concentration measurements and samples taken at flexible times.

Despite these advantages, the implementation of Bayesian methods into healthcare settings has been limited, even though some national guidelines recommend this approach mainly for antimicrobial drugs with easily available TDM, i.e., non-BL antibiotics [66]. Several potential barriers to the widespread implementation of Bayesian methods in clinical practice have been identified. The lack of PK expertise and easy-to-use dosing tools to support clinicians in better tailoring antimicrobial dosing are likely to explain the poor implementation of Bayesian methods. On the other end, the need to integrate individual conditions to increase dosing precision has grown. To maximize the success of these efforts and to further facilitate their implementation into clinical practice, there is a clear need for using friendly software tools.

### 5.2. Clinical Data Supporting Antimicrobial Dosing Software Use 

Available clinical trials supporting BL dosing software tools are summarized in Table 3. 

A growing body of evidence indicates, especially in critically ill patients, that beta-lactam TDM-guided dosing and MIPD may maximize efficacy. Felton et al. analyzed the fitness of BestDose^TM^ software to estimate each individual’s pharmacokinetics by comparison of the observed-versus-predicted piperacillin concentration after 24 h of therapy and its ability to accurately predict piperacillin dosing from the observed piperacillin concentrations [57]. They found that the dose optimization software predicted a mean piperacillin dosage of 4.02 g in the eight patients administered piperacillin doses of 4.00 g when at least two observed piperacillin concentrations were available. Moreover, linear regression of the observed-versus-predicted piperacillin concentrations for the eight individuals including in the study demonstrated an r^2^ of >0.89. Heil et al. demonstrated that the use of the PK/PD-based antibiotic dosing calculator- ID-ODS resulted in the probability of target attainment of 98% in 50 severely ill patients treated with cefepime, meropenem or piperacillin-tazobactam [67]. Similarly, Chiriac et al. have shown that combining the application of dosing software and consecutive TDM increases therapeutic drug exposure of piperacillin in patients with sepsis and septic shock [68]. Additionally, in 12% of patients with excessive and potentially toxic piperacillin concentrations, subsequent TDM-guided dosing adjustments enabled the adequate reduction of piperacillin dosing and exposure [68]. In this subtherapeutic group, higher mortality was observed compared to the therapeutic group for comparable severity scores. Even though no strong conclusion may be drawn based on these clinical outcome findings, these results confirm the need for dose BL dose optimization with two objectives: achieving therapeutic exposure and avoiding potentially harmful BL concentrations.

However, the DOLPHIN trial, a recent multicenter trial comparing BL TDM in 388 adult ICUs using MIPD and pharmacometrics modeling to TDM according to usual care, did not improve outcomes such as length of stay or mortality [69]. Unexpectedly, the rate of target attainment in the MIPD was quite low (ranging from 55.6% to 71.4%) [70]. This study highlights some important considerations while using MIPD dose optimization:-First, the importance of early, appropriate and adequately dosed empiric antimicrobials as TDM can only be applied after empiric antimicrobial dose selection, and initial dosing may be more predictive of meaningful outcomes than TDM assisted maintenance dosing only.-Second, the choice of the dosing software and the PK models is a key determinant for antimicrobial MIPD-guided optimization success. The PK/PD dosing software used in the DOLPHIN trial—InsightRx^®^- although registered as a CE-labeled medical device with published embedded PK models, failed to predict adequate BL and ciprofloxacin doses in ICU patients. As previously reported, the external evaluation of published population PK models may lead to poor predictive performance when applying to a cohort of ICU patients different from the one used to develop the PK models as shown for meropenem [71,72]. Thus, it is necessary to perform a fit-for-purpose evaluation of the models to assess their predictive performance in the MIPD setting they are intended to be used before any implementation.-Third, only 61% of patients in the MIPD group had a second TDM sampling. As duration of therapy was short (median duration of therapy: 4 days in MIPD group vs. 3.5 in standard dosing group) together with delayed dosing optimization due to informed consent, the benefit of MIPD-guided dosing on clinical outcomes may be limited by a restricted interventional period.

### 5.3. Barriers to the Widespread Use of Dosing Software That Need to Be Overcome

Before MIPD becomes common clinical reality, several issues must be addressed [73].

-First, in the near future, the regulatory framework for antimicrobial dosing software tools needs to be reinforced. Software developers may be required to register their dosing software with relevant regulatory bodies before health services are able to incorporate the technology.-Second, the difficulty of MIPD use for untrained healthcare providers questions the integration of specialized pharmacists or pharmacologists into clinical teams. Indeed, to drive adoption of these tools in clinical practice, it is essential to provide proper education of the intended end-user. The lack of dedicated time for practitioners to use these tools on a larger scale together with a lack of knowledge about the reliability of software outputs, as well as understanding of how the software works, have been identified as influencing trust in dose-prediction software [74]. Implementation of MIPD into dosing advisory service deserves consideration to facilitate translation of Bayesian forecasting dosing recommendations into clinical practice as already demonstrated for vancomycin [74].-Third, the ability to integrate dose-prediction software within existing hospital electronic medication management systems helps minimizing the need for prescribers to input data and is a key aspect considered by prescribers when deciding to accept software. Providing clinicians with quality-assured user-friendly decision support tools available in application form for personal mobile devices, integrated into electronic hospital record (EHR) prescribing software, is of paramount importance for the widespread implementation of MIPD.-Fourth, cost-effectiveness of using BL dosing software must be demonstrated. Precision dosing may require additional costs initially for analysis of drug concentration or other biomarkers that provide information necessary for optimal dose selection. These analyses, though theoretically cost-effective, may require a learning curve for clinicians before expenditures are reduced in clinical practice. Another cost associated with precision dosing is the integration of drug dosing software into EHRs. Although favorable cost outcomes from using dosing software for non-BL antibiotics have been reported, the DOLPHIN trial found incremental costs of EUR 5312 with an average decrease in 6 months QALY of 0.03 (range −0.5 to 0.5) in MIPD compared to the standard dosing [69,75].

## 6. Beta-Lactam Therapeutic Drug Monitoring for the Critically Ill Patient: Time to Focus on Rationale of TDM Use in Special ICU Populations?

Clinical awareness that dosing requires fine-tuning to achieve the desired effects has grown in special critically ill populations. 

Obese patients are sometimes excluded from the screening phase, even though these are patients for whom the optimization of drug doses is challenging on a daily basis, especially since their prevalence is increasing dramatically [76]. Therapeutic failures in obese patients are common and responsible for longer hospital stays even in less serious patients [77]. PK variability in obese patients is due to alterations in Vd and clearance resulting from increased adipose tissue and lean muscle mass. A study focusing on piperacillin-tazobactam TDM in severely obese ICU patients emphasized the increased risk of underexposure and high interindividual variability in this population [78].

Renal failure leads to a risk of toxicity by overdosing, proving the interest of anticipating increasing serum concentrations [47]. Although there is extremely high variability of BL exposure in patients with renal failure or those undergoing RRT supports TDM and dose individualization, few studies have investigated whether TDM and subsequent dose adjustment could reduce such variability and improve BL exposure [79]. To date, no randomized study assessing TDM interventions in patients undergoing RRT has been conducted, and the optimal dose adjustment approach (empirical dosing, nomograms or dosing software) remains to be determined. MIPD dosing and innovative approaches such as the integrated dialysis pharmacometrics model that incorporates all RRT samples (pre- and post-filter plasma and effluent), RRT types and settings and drug physiochemical properties to provide quantitative insight into drug clearance in patients receiving RRT should be considered in future studies [80].

On the other hand, critically ill patients exhibiting augmented renal clearance defined as CrCl above 130 mL/min/1.73 m^2^ are more likely to experience BL underexposure potentially leading to clinical failure when BL dose are not adjusted [81]. A single-center prospective study focusing on unbound BL concentrations in ICU patients highlighted that one third of dosing regimens required a dose increase to achieve therapeutic exposure and showed an increased risk of failure in patients with ARC (OR 2.47) [21,82]. In 215 septic patients with ARC (median CrCl = 178 mL/min), Jacobs et al. found that subtherapeutic concentrations of BL varied from 36% of the cohort when considering *Enterobacterales* breakpoint up to 55% for *Pseudomonas aeruginosa.* The proportion of inadequate BL plasma concentrations significantly increases (up to 100% for piperacillin) in the upper range of measured CrCl (240–300 mL/min). However, as weak correlations between measured CrCl and BL plasma concentrations and between measured CrCl and BL clearances were observed in this study, dosing adjustments based on measured CrCl only are unlikely to achieve PK/PD targets. Under these circumstances, TDM should be strongly encouraged to guide BL drug dosing as off-label dosing is often necessary to achieve therapeutic concentrations. Interestingly, the package insert of recently released antibiotics integrates ARC dosing recommendations, as seen for cefiderocol. MIPD may also be valuable in this population but has not been investigated yet. Post hoc analyses of the DOLPHIN trial are currently underway in such population and may inform future trials. 

Finally, another challenging situation for BL dosing optimization is infection with less susceptible pathogens requiring higher BL exposure to achieve PK/PD targets. In a recent single center retrospective study, meropenem dose optimization was successfully achieved by dosing adjustment based on real-time TDM for infections caused by KPC with a meropenem MIC ≤ 64 mg/L [83]. Besides, as rapid antimicrobial susceptibility testing is currently being developed, TDM-based dosing adjustments considering true and rapidly available MICs may further improve BL therapy at the early phase of infections with difficult-to-treat pathogens.

## 7. Future Perspectives

Further well-designed prospective clinical trials will be required to determine the benefit of precision dosing and will be crucial to put the effort of TDM into cost effectiveness. In the near future, a multicenter randomized controlled superiority trial aiming to assess the effectiveness and safety of data driven automated antimicrobial dosing advice and a study combining the use of a Bayesian model with rapid molecular diagnostic test in order to offer an optimal personalized care will provide new insights into optimized TDM and MIPD benefit [84,85].

Artificial intelligence (AI) embedded into dosing software is a promising strategy [86]. AI software uses reinforcement learning to provide recommendations on dose adaptation. AI software analyzes large patient databases to determine interventions that may influence patient outcome. AI algorithms and machine learning allow to continuously refine existing models based on the specific population, thus maximizing the probability to achieve PK/PD target. 

## 8. Conclusions

Beta-lactam dosing optimization is a cornerstone of critically ill patient management. Therapeutic drug monitoring of BL antibiotics and the ability to individualize dosing regimens remains a promising approach with great potential to impact selective patient populations at high risk of inadequate BL exposure. For several reasons, current studies evaluating BL dosing optimization fail to demonstrate a decline in mortality. The combination of Beta-lactam dosing optimization with rapid molecular diagnostic testing to offer an optimal personalized care could be an interesting approach when designing further studies. The benefit of MIPD remains to be determined from safety and cost point of views. In the meantime, efforts must be pursued on an available, simple-to-operate and cost-effective TDM program that provides a short turnaround time in combination with reliable and easy-to-interpret results in the critical care setting.

## Figures and Tables

**Figure 1 antibiotics-12-00681-f001:**
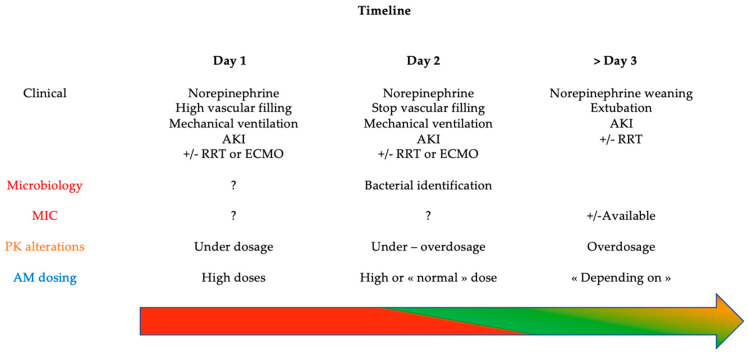
Timeline of PK/PD alterations in critically ill patients with septic shock. AKI: acute kidney injury, RRT: renal replacement therapy, ECMO: extracorporeal membrane oxygenation, VF: vascular filling, MIC: minimal inhibitory concentration, PK: pharmacokinetics, AM: antimicrobial; Arrow with color: level of inadequate BL concentration risk: red = very high, green = low, orange = variable.

**Figure 2 antibiotics-12-00681-f002:**
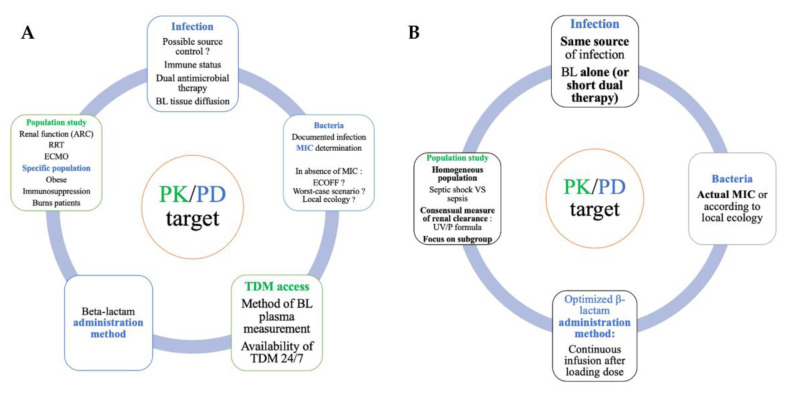
Keypoints to consider when constructing future studies on PK/PD target attainment ((**A**), upper panel) and suggested five tips ((**B**), lower panel). ARC: augmented renal clearance, RRT: renal replacement therapy, ECMO: extracorporeal membrane oxygenation, BL: beta-lactam, MIC: minimal inhibitory concentration, TDM: therapeutic drug monitoring, PK/PD: pharmacokinetics/pharmacodynamics.

**Figure 3 antibiotics-12-00681-f003:**
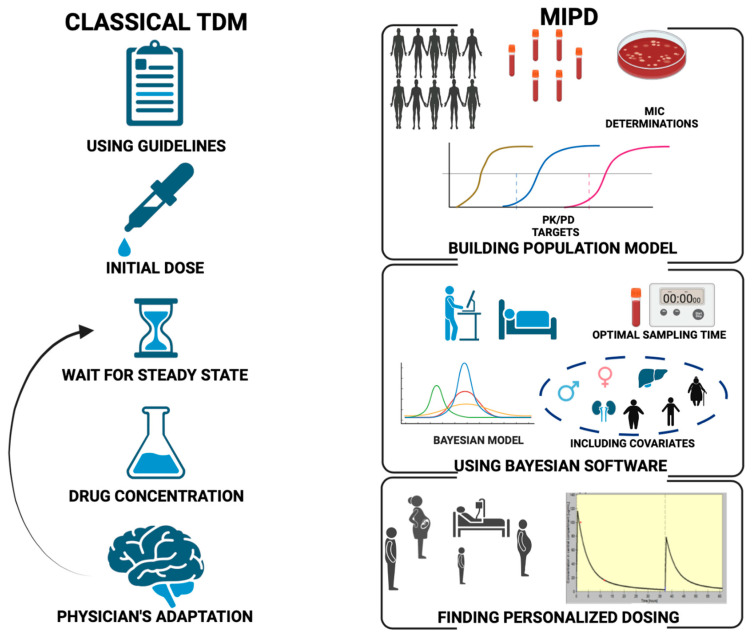
Comparison of the traditional TDM (**left**) and MIPD (**right**) workflows. MIC: minimum inhibitory concentration; MIPD: model-informed precision dosing; PD: pharmacodynamic; PK: pharmacokinetic; TDM: therapeutic drug monitoring.

**Table 1 antibiotics-12-00681-t001:** Barriers to counter for a better TDM adherence.

Laboratory Level		Clinician and Pharmacist Level
Microbiology	Pharmacology	
Consensus of MIC measurement method To accelerate MIC determination To accelerate bacteria identification	Consensus of BL measure method (total vs. free concentration) To accelerate results obtention To develop method of tissue concentration determination	Consensus on PK/PD target to adopt To define the therapeutic range ⇔ make the result easier to interpret for non-TDM expert To identify populations that would most benefit from TDM
Main objective: availability of the MIC result	Main objective: availability of the TDM result	Main objective: interpretation of TDM result

MIC: minimal inhibitory concentration, TDM: therapeutic drug monitoring, PK/PD: pharmacokinetics/pharmacodynamics.

**Table 2 antibiotics-12-00681-t002:** Characteristics of the four most recent PK/PD target randomized control studies.

Study	Population/Beta-Lactam	Group	PK/PD Target	Microbiology	TDM/Sampling	Outcomes
De Waele et al., 2014 [35]	**-n = 41** Septic patient with normal renal function ^1^ (GFR > 80 mL/min) APACHE II score 18 (13–24) Day 1 SOFA score 5 (2–6) **-Meropenem** (LD 1 g followed by 1 g/8 h) **Piperacillin/tazobactam** (LD 4 g followed by 4 g/6 h) Extended perfusion	-Intervention: TDM-guided dosing Dose adjustment protocol -Control: conventional dosing	**-100%****ƒ****T**_MIC>4_ Within the first 72 h -MIC: ECOFF wild-type Pseudomonas aeruginosa (16 mg/L PIP/TAZ and 2 mg/l MEM)	-Documented infection: 66% -n = 6 for *P. aeruginosa* Median MIC 2 (1.5–8) mg/L for PTZ MIC 0.125 (0.125–0.690) mg/L for MEM	-Daily TDM in both group but the control group was blinded of the results -Analytic method of beta-lactam well described -Total concentration of antibiotic dosed Intervention group: need for dose optimization n = 16 (76%)	-TDM allowed a high median 100% ƒT_MIC>4_ at day 3 -28-day mortality: Control n = 5 (25%) Intervention n = 3 (14%)
Sime et al., 2015 [33]	**-n = 32** Hematological malignancies Febrile neutropenia Normal renal function ^2^ (creat clearance > 75 mL/min/1.73 m^2^) Various dual therapy (mostly gentamicin) **-Piperacillin/tazobactam** 4.5 g/8 h Intermittent infusion	-Intervention: Daily TDM and protocol of dose adjustment -Control: daily TDM but no dose adjustment	**-100%****ƒ****T** > **MIC** Within the first 72 h -MIC: actual or ECOFF wild-type Pseudomonas aeruginosa (16 mg/L) and enterobacterales in negative culture	-Documented infection: 41% -Mostly enterobacterales no *P. aeruginosa* MIC not reported	-2 blood samples/day or after any dose change: 50% of the dosing interval and 15 min prior to the next dose -No detail of analytic method -Total concentration of antibiotic dose and estimated free based on 30% protein binding	-Improvement of the rate of PK/PD target attainment in the intervention group for the second and third TDM -No difference in time to neutrophil recovery or fever resolution
Fournier et al., 2018 [34]	**-n = 38** Burn patients 73 episodes of infection mostly pneumonia 61% of appropriate initial antibiotic treatment (among the 38% non-appropriate, most of underdosing) **-Various beta-lactams** Intermittent administration and then extended perfusion	-Intervention: real-time daily TDM and online adaptation protocol -Control: usual care, no access to TDM results	**-C**_**min**_**>****MIC** actual or ECOFF from local ecology	-Documented infection: 85% -Mostly *P. aeruginosa* and *S. aureus*	-244 TDM measures -Analytic method of beta-lactam well described. -Total concentration of antibiotic dosed and estimated free fraction based on published data	-C_min_ level target higher in the intervention group (74%) vs. 56.5% in the control group (*p* = 0.018) -No difference in infection outcomes
Hagel et al., 2022 [38]	**-n = 242** General ICU population APACHE II score 23.2 ± 6.7 Day 1 SOFA score 12.1 ± 2.8 **-Piperacillin/tazobactam** (LD 4.5 g followed by continuous infusion of 13.5 g	-Intervention: real-time daily TDM no algorithm for dose adjustment -Control: usual care, no access to TDM results, daily dose adjustment according to renal function assessed by ^1^GFR	**-100%****ƒ****T**_**MIC****>**4_ -MIC actual or ECOFF wild-type Pseudomonas aeruginosa (16 mg/L) for empirical therapy	-Documented infection: 65% -Mostly *E. coli*, *K. pneumoniae* and *S. aureus* -Actual MIC ≤ 4 mg/L (80%)	-1179 TDM measures mostly performed on day 1 -On-site measurements of total piperacillin concentration	-No significant beneficial effect of TDM with regard to the 10-day mean total SOFA score -Less mortality in the TDM group of 4.2% without statistical significance -More PK/PD target attainment in the TDM group

GFR: glomerular filtration rate, ICU: intensive care unit, SOFA: Sequential Organ Failure Assessment, LD: loading dose, MIC: minimal inhibitory concentration, TDM: therapeutic drug monitoring, PK/PD: pharmacokinetics/pharmacodynamics. ^1^ GFR according to CKD-EPI equation, ^2^ Creatinine clearance according to Cockroft-Gault formula. For piperacillin/tazobactam, only the piperacillin was measured.

**Table 3 antibiotics-12-00681-t003:** Characteristics of the four clinical trials reporting BL dosing software clinical performance in critically ill patients.

Study	Population/Beta-Lactam	Study Design	PK/PD Target	Microbiology	Software/Sampling	Outcomes
Felton et al., 2014 [57]	**-n = 40** -Adult ICU patients with ventilator-associated pneumonia -Piperacillin	Prospective population PK study	**100%** ƒ**T**_MIC>1_	Not available	**-Dosing software: BestDose**^TM^ -Time of sampling: 24 h after start of therapy	-Linear regression of the observed-versus-predicted piperacillin concentrations demonstrated an r^2^ of >0.89 -Good predictive performance for prescribed dose
Heil et al., 2018 [67]	**-n = 49** -General adult ICU population, Mean SOFA score: 6 (3.3) -Cefepime, Meropenem, and Piperacillin-Tazobactam	Prospective observational study	ƒ**T**_MIC>1_ 50% for piperacillin, 40% for meropenem, 60% for cefepime	*P. aeruginosa* 29% mostly enterobacterales	**-Dosing software: ID-ODS**^TM^ -Time of sampling: 50% of the dosing interval and trough	-Target attainment: 98%
Chiriac et al., 2021 [68]	**-n = 179** -General adult ICU population RRT 29%/Septic shock 30%/Pneumonia 50%/peritonitis 23%Mean SOFA score: 6 (6) -Piperacillin	Retrospective study	**100%** ƒ**T**_MIC>2–4_ MIC ECOFF for Pseudomonas aeruginosa 16 mg/L	*P. aeruginosa* 6% mostly enterobacterales	**-Dosing software: CADDy** -Time of sampling: 24–48 h after start of therapy	-Target attainment: 40% -TDM-guided dose adjustments significantly enhanced therapeutic exposure to 65%, and significantly reduced piperacillin concentrations> 96 mg/L to 5%.
Ewoldt et al., 2022 [69]	**-n = 388** -General adult ICU population APACHE IV score: 70 IQR (51–90)/SOFA score 8 (5–10.3) -Various Beta-lactams (mostly ceftriaxone, cefuroxime and meropenem) (+ciprofloxacin 30% of the total cohort)	RCT Intervention: MIPD with TDM and algorithm for dose adjustment Control: Standard dose regimens and adjustment according to local guidelines TDM in both group	**100%** ƒ**T**_MIC>4_ MIC ECOFF for the expected pathogen Piperacillin: MIC of 16 mg/L Meropenem: MIC of 2 mg/L	Documented infection: 53% (n = 206) Pathogens: Pseudomonas spp. 7% mostly enterobacterales	**-Dosing software: InsightRx**^TM^ -Time of sampling: T1 (first antibiotic sampling), day 3 and day 5 Median time of first sample: 19.6 h (intervention group) Dose adjustment: 37.6% at T1 (intervention group) and 11% (control group) On-site measure of total concentration except for highly bound antibiotic (ceftriaxone, flucloxacillin) where unbound fraction was measured	-Target attainment T1 55.6% (intervention)–61% (control) -No additional benefice of MIPD of beta-lactams on ICU length of stay, mortality, target attainment

RCT: randomized controlled trial, TDM: therapeutic drug monitoring, MIC: minimal inhibitory concentration, MIPD: model informed precision dosing, PK/PD: pharmacokinetics/pharmacodynamics, ICU: intensive care unit, SOFA: Sequential Organ Failure Assessment.

## Data Availability

No new data were created or analyzed in this study. Data sharing is not applicable to this article.

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
