# Peer review of "The Current Status and Future Perspectives of Beta-Lactam Therapeutic Drug Monitoring in Critically Ill Patients"

_antibiotics, 2023, doi:10.3390/antibiotics12040681_

Round 1
Reviewer 1 Report
Thank you for reviewing the paper from Novy E et al. entitled “Beta-lactam therapeutic drug monitoring for the critically ill patient: time to think differently”. I read the paper with many interests. To summarize, the authors present important insights to the topic of antibiotic TDM (especially BL antibiotics) in critical ill patients, e.g., its present state, its importance in clinical practice, and pitfalls. The authors also critical review present studies and offer a point-to-point optimizations strategy for further studies as well as critical discuss the importance of software tools as tool for therapeutic optimization of antibiotic dosage in critical ill patient. The review is well understandable and addresses a very important topic and I have no further comments.
Author Response
We thank reviewer 1 for his/her positive comments on our paper
Reviewer 2 Report
The manuscript, entitled "Beta-lactam therapeutic drug monitoring for the critically ill patient: time to think differently", is a review devoted to the problems of practical application of beta-lactam therapeutic drugs. The topic of the article is extremely important, since the introduction of new drugs is very often delayed for a variety of reasons, so reviews that collect information about problems and ways to solve them help promote new therapy. The authors have studied this topic in some detail, their ideas and suggestions are valuable. The conclusions are not formulated specifically enough. The text is logical and structured, but there are a number of shortcomings. It is proposed to accept the article after minor changes.
1) Page 1, line 31; page 2, line 63. A colon is written instead of a dot.
2) Page 4, line 190; Page 11, line 103. The signature to Figure 1 and Figure 1 itself must be placed on the same page. This also applies to Figure 3.
3) Page 5. There is too much free space not occupied by the text.
4) Page 10. All the signatures below should be placed on the same page with the figure.
5) Page 13, lines 171-173. It is advisable to align this place, visually the text is not in order.
6) Page 12, line 120 and so on. Some headings are not numbered.
7) Extra dots are placed in the places of references to literary sources.
8) The titles of the chapters repeat the title of the article. This does not seem to be the most successful way to organize the text. It is advisable to make the titles of chapters and sub-chapters more specific and shorter.
9) The conclusion does not reflect all the conclusions received in the text.
Author Response
We thank reviewer 2 for his/her valuable comments on our paper.
1) Page 1, line 31; page 2, line 63. A colon is written instead of a dot.
Answer: The correction has been made.
2) Page 4, line 190; Page 11, line 103. The signature to Figure 1 and Figure 1 itself must be placed on the same page. This also applies to Figure 3.
Answer: The signatures and the figures are now on the same page in the revised manuscript.
3) Page 5. There is too much free space not occupied by the text.
Answer: Free space has been removed in the revised manuscript.
4) Page 10. All the signatures below should be placed on the same page with the figure.
Answer: This correction has been made;
5) Page 13, lines 171-173. It is advisable to align this place, visually the text is not in order.
Answer: Thank you for this remark. We have now edited this paragraph accordingly.
6) Page 12, line 120 and so on. Some headings are not numbered.
Answer: All the headings are now numbered in the revised manuscript.
7) Extra dots are placed in the places of references to literary sources.
Answer: we have double checked all the references.
8) The titles of the chapters repeat the title of the article. This does not seem to be the most successful way to organize the text. It is advisable to make the titles of chapters and sub-chapters more specific and shorter.
Answer: Thank you for this comment. We have now shortened the titles of the chapters and make them more focused.
9) The conclusion does not reflect all the conclusions received in the text.
Answer: Thank you. We have now edited the conclusion in order to better reflect the points highlighted in the narrative review.
Reviewer 3 Report
This review article provides a good summary of BL TDM in ICU patients.
1. Overall, the readability of tables and figures must be improved.
2. The abbreviation appears in the first sentence of the text. Please explain the abbreviation at the first appearance.
3. The main title of 3~6 sections is the same: Beta-lactam therapeutic drug monitoring in the critically ill patient. Please unify the 3~6 sections and classify them into subheadings.
4. Title may need revision. It would be better to change it to be similar to "The current status and future prospects of BL TDM in critically ill patients". "time to think differently" is non-informative.
Author Response
We would like to thank Reviewer 3 for his/her valuable remarks.
1. Overall, the readability of tables and figures must be improved.
Answer: Thank you for this comment. We have edited the tables and figures in order to improve the readability in the revised manuscript.
2. The abbreviation appears in the first sentence of the text. Please explain the abbreviation at the first appearance.
Answer: Sorry for this mistake. We have now added the full text and edited abbreviation as recommended.
3. The main title of 3~6 sections is the same: Beta-lactam therapeutic drug monitoring in the critically ill patient. Please unify the 3~6 sections and classify them into subheadings.
Answer: Thank you for this comment. We have now changed the titles of the chapters as suggested by Reviewer 2 and made them shorter and more specific.
4. Title may need revision. It would be better to change it to be similar to "The current status and future prospects of BL TDM in critically ill patients". "time to think differently" is non-informative.
Answer: We have now changed the title as suggested for: "The current status and future perspectives of beta-lactam therapeutic drug monitoring in critically ill patients."